

# Upper extremity kinematics: development of a quantitative measure of impairment severity and dissimilarity after stroke

Khadija F. Zaidi[1] and Michelle Harris-Love[2,3]

[1] Department of Bioengineering, George Mason University, Fairfax, United States
[2] University of Colorado, Anschutz Medical Campus, Aurora, Colorado, United States
[3] Medstar National Rehabilitation Hospital, Washington, District of Columbia, United States of America

## ABSTRACT

**Background:** Strokes are a leading cause of disability worldwide, with many survivors experiencing difficulty in recovering upper extremity movement, particularly hand function and grasping ability. There is currently no objective measure of movement quality, and without it, rehabilitative interventions remain at best informed estimations of the underlying neural structures' response to produce movement. In this article, we utilize a novel modification to Procrustean distance to quantify curve dissimilarity and propose the Reach Severity and Dissimilarity Index (RSDI) as an objective measure of motor deficits.

**Methods:** All experiments took place at the Medstar National Rehabilitation Hospital; persons with stroke were recruited from the hospital patient population. Using Fugl-Meyer (FM) scores and reach capacities, stroke survivors were placed in either mild or severe impairment groups. Individuals completed sets of reach-to-target tasks to extrapolate kinematic metrics describing motor performance. The Procrustes method of statistical shape analysis was modified to identify reaching sub-movements that were congruous to able-bodied sub-movements.

**Findings:** Movement initiation proceeds comparably to the reference curve in both two- and three-dimensional representations of mild impairment movement. There were significant effects of the location of congruent segments between subject and reference curves, mean velocities, peak roll angle, and target error. These metrics were used to calculate a preliminary RSDI score with severity and dissimilarity sub-scores, and subjects were reclassified in terms of rehabilitation goals as Speed Emphasis, Strength Emphasis, and Combined Emphasis.

**Interpretation:** The modified Procrustes method shows promise in identifying disruptions in movement and monitoring recovery without adding to patient or clinician burden. The proposed RSDI score can be adapted and expanded to other functional movements and used as an objective clinical tool. By reducing the impact of stroke on disability, there is a significant potential to improve quality of life through individualized rehabilitation.

Corresponding author
Khadija F. Zaidi, szaidi8@gmu.edu

## INTRODUCTION

Strokes represent one of the leading causes of disability worldwide. A total of 65% of stroke survivors experience some difficulty in recovering the ability to reach (*Krauth et al., 2019*; *Mesquita et al., 2019*, *2018*), with more severe impairments featuring a loss of hand function and ability to grasp (*de los Reyes-Guzmán et al., 2014*; *Reissner et al., 2019*; *Niechwiej-Szwedo, Nouredanesh & Tung, 2021*). At 6 months post-stroke, many continue to experience some degree of upper extremity hemiparesis. This unilateral impairment of the paretic limb impacts functional reaching and is a major contributor to stroke-related disability (*Mohapatra et al., 2016*).

Early signs of motor control interruption include paralysis, reduced reflexes, and inability to produce resistance to perturbations (*Cacioppo et al., 2020*; *Sing, Orozco & Smith, 2013*). Symptoms arising during the chronic post-stroke recovery phase may include increased reflex activity or spasticity (*Elliott et al., 2020*; *Harris-Love, 2012*). Compensatory movements may also arise in lieu of true recovery, such as extending the trunk to reach a target at arm's length due to decreased joint range of motion (*Saes et al., 2022*). Stroke severity can significantly impact the type and amount of deficits experienced by an individual and the efficacy of particular rehabilitative strategies (*Cirstea & Levin, 2000*). While mechanisms of arm recovery have been studied after mild functional impairments (*Morel, Ulbrich & Gail, 2017*; *Schwarz et al., 2021*), there are few effective treatments for the large portion of the stroke population with more severe impairments. An objective measure of severity and the nature of deficits is of interest in creating individualized rehabilitation plans (*Wolff et al., 2023*; *Yang et al., 2018*).

Three-dimensional kinematic analyses provide objective methods to characterize movement subsequent to stroke (*Cai et al., 2019*; *Jarque-Bou et al., 2020*; *Schwarz et al., 2019*; *Wolff et al., 2022*). Kinematics of the upper extremity obtained through motion capture and 3D positional data can provide more sensitive tools to objectively assess individual motor function after stroke (*Scano, Molteni & Molinari Tosatti, 2019*; *Ozturk et al., 2016*; *Jaspers et al., 2011a*). Active and passive visual markers, electromagnetic sensors, and inertial sensors have been used extensively for human movement analysis and can provide metrics such as movement speed, movement smoothness, joint angles, and limb orientation from position data (*Collins et al., 2018*; *Ueyama, 2021*).

Currently, there is no consensus on the most appropriate tasks or variables to provide a global description of upper extremity movement (*McCrea & Eng, 2005*; *Corona et al., 2018*; *van Andel et al., 2008*). With significant variability between individuals, clinicians use measures such as the upper extremity Fugel-Meyer scale to subjectively describe movement capability (*Singer & Garcia-Vega, 2017*; *Woytowicz et al., 2017*). Without an objective measure of movement quality, rehabilitative interventions are at best informed estimations of how the underlying neural structures will respond and produce movement. Subjective clinical scores cannot identify where during movement a deficit occurs and what that might suggest as the best rehabilitative plan (*Roberts & Lawrence, 2019*; *Roberts, 2020*; *Priot et al., 2020*). Subjective scales also cannot efficiently monitor changes in impairment severity and dissimilarity over time.

In this article, we propose a modified Procrustes analysis method applied to groups of persons with stroke, differentiated by movement severity. Utilizing upper extremity endpoint data from these two groups, this method was used to identify movement behaviors and metrics that differentiate the mild and severe impairment groups. Finally, this study includes a preliminary severity and dissimilarity score of upper extremity movement that draws inspiration from scores such as the Gait Profile Score (GPS) (*Baker et al., 2009*), or Gait Deviation Index (GDI) (*Schwartz & Rozumalski, 2008*). The GPS evaluates overall gait pathology severity based solely on kinematic data for a given individual, while the GDI identifies how much an individual's gait features deviate from a reference set of able-bodied data. A single measure of the overall quality of an upper extremity movement, overall severity, and dissimilarity from reference data would be of interest in informing clinical decisions.

The article is proceeds as follows:

- Validity of utilizing Procrustean distance in upper extremity analysis,
- Objectives and hypotheses of applying a modified Procrustes analysis to endpoint data,
- This study's inclusion and exclusion criteria for persons with stroke,
- Clinical measures used to classify patients into mild and severe impairment groups,
- Description of the experimental protocol and study methodology,
- Definitions of kinematic metrics included in the data analysis,
- Statistical tests performed to identify significant differences between subject groups,
- Resulting quantitative measures of severity and dissimilarity that inform the proposed Reach Severity and Dissimilarity Index (RSDI).

## PROCRUSTEAN DISTANCE IN MOVEMENT ANALYSIS

In mathematics, the Euclidean distance between two points is the length of a line drawn between them. Root-mean-square error (RMSE) is another method of quantifying how much one set of data differs from a reference set. Both Euclidean distance and RMSE have been used to construct measures of movement quality in the lower limb (*Baker et al., 2009*; *Karamanidis, Arampatzis & Brüggemann, 2016*) and the upper limb (*Cerveri et al., 2007*; *Dehbandi et al., 2016*; *Riad et al., 2011*). Additionally, principle component analysis (PCA) is commonly used to simplify the interdependent data that is necessary to represent participating limb segments and joints, task requirements, and environmental constraints that produce any particular movement (*Guzik-Kopyto et al., 2022*). Clinical decisions can then be based on an interpretation of the complex data. The validity of scores generated by quantifying differences between mean reference data and paretic movement data has been established in the field of rehabilitation (*Jaspers et al., 2011b*; *Hill, Mong & Vo, 2022*).

Procrustes analysis is another such psychometric method of quantifying difference or dissimilarity between two sets of data (*Kendall, 1989*). Procrustes distance has recently garnered attention as a metric in both gait (*Rida, Almaadeed & Almaadeed, 2019*; *Sehairi, Chouireb & Meunier, 2018*; *Anwary, Yu & Vassallo, 2019*) and upper extremity studies (*Passos et al., 2023*; *Saenen, Orban de Xivry & Verheyden, 2022*; *Wong et al., 2019*; *Passos,*

*Campos & Diniz, 2017*). Procrustes analysis quantifies the similarity of shape between two matrix sets and provides the linear transformation that would allow one curve to best conform to the other. More specifically, the Procrustes method compares each ith element of the subject curve to the ith element of the reference curve. This method generates a scaling factor b, an orthogonal rotation/reflection matrix T, a translation matrix C, and a Procrustes distance d. Computing the Procrustes distance presents an interesting advantage in quantifying subject performance. Additionally, the scaling factor b can indicate a prolonged or truncated movement, while the ability to compare a reflected curve can allow comparison of right and left limb movements to the same reference curve (*Seber, 2009*; *Bookstein, 1997*). In addition to discrete kinematic landmarks, the variability across an entire movement can be assessed in order to extrapolate a subjective and sensitive representation of upper limb movement.

In order to support the proposed RSDI score, we quantitatively identified segments of the forward-reaching movement that showed the least deviation when compared to a reference curve representing stereotypical able-bodied reaching behavior. These segments of the movement were characterized not by the magnitude of discrete kinematic metrics but rather by when they occur relative to those metrics and during the overall movement. We hypothesize that subjects with mild impairment will exhibit initial acceleration behaviors that are analogous to healthy movement, while subjects with more severe impairment will not exhibit any congruous segments of movement and therefore result in higher severity and deviation scores. Further, it is expected subjects with severe impairment will demonstrate a diminished ability to refine movement through less variability in endpoint orientation. This study suggests the specific sub-movements, in cases of mild and severe impairment, that remain congruous to healthy movement can allow quantification of impairment severity and inform targets for rehabilitation.

## METHODS

### Inclusion and exclusion criteria

All participants were recruited from the MedStar National Rehabilitation Hospital stroke patient population. Patients' stroke diagnoses were confirmed *via* magnetic resonance imaging (MRI). This protocol was approved by the Medstar Rehabilitation Research Institutional Review Board under protocol number (947339-3).

Persons with stroke that were (1) at least eighteen years of age, (2) able to complete a reach-to-target task, (3) able to consent to the study and comprehend instructions, and (4) six or more months post thromboembolic non-hemorrhagic hemispheric or hemorrhagic hemispheric strokes were recruited for this study.

Potential subjects were excluded if (1) they were less than 18 years of age, (2) stroke occurred less than 6 months before participation or affected both hemispheres, (3) stroke involved the cerebellum, brainstem, or did not spare primary motor and dorsal premotor cortices, (4) there was a history of craniotomy, neurological disorders (other than stroke), cardiovascular disease, or active cancer or renal disease, (5) there was a history of orthopedic injury or disorder affecting shoulder or elbow function, or (6) they had had a seizure or taken anti-seizure medication in the past 2 years.

**Table 1 Mild impairment stroke survivor demographics.**

| Sub | M/F | Age (years) | Time since stroke (months) | Paretic arm | Dominant affected | Reach capacity (cm) | UEFM (max score = 66) |
|---|---|---|---|---|---|---|---|
| 1 | M | 62 | 107 | R | Yes | 47.4 | 59 |
| 2 | M | 64 | 72 | R | Yes | 46.5 | 51 |
| 3 | M | 44 | 14 | L | No | 43.0 | 46 |
| 4 | M | 64 | 14 | L | No | 36.3 | 43 |
| 5 | M | 54 | 13 | R | Yes | 40.8 | 42 |
| 6 | M | 59 | 68 | L | No | 27.2 | 54 |
| 7 | F | 57 | 22 | L | No | 20.5 | 64 |
| 8 | M | 60 | 48 | L | No | 29.2 | 63 |
| 9 | M | 44 | 42 | L | Yes | 41.5 | 64 |
| 10 | M | 73 | 8 | R | Yes | 19.6 | 50 |
| 11 | M | 77 | 55 | R | Yes | 44.8 | 43 |
| 12 | F | 74 | 7 | R | Yes | 32.1 | 61 |
| 13 | M | 65 | 78 | L | No | 29.2 | 53 |
| 14 | F | 59 | 20 | R | Yes | 37.5 | 39 |
| 15 | F | 71 | 17 | L | No | 37.8 | 54 |
| | 11M | 61.8 | 39 | 7R | 9Y | 35.6 | 52.4 |
| | 4F | ±9.8 | ±31 | 8L | 6N | ±8.9 | ±8.5 |

**Note:**
UEFM—upper extremity Fugl-Meyer.

Subjects underwent examinations by clinicians prior to study recruitment (*Folstein, Folstein & McHugh, 1975*) to ensure ability to consent to all sections of the study and complete tasks as instructed. As this study features a functional reaching task for the upper extremity only, assessment of recovery was limited to the upper extremity motor function section of the Fugl-Meyer Assessment. The upper extremity Fugl-Meyer (UEFM) test was used as a criterion for classifying post-stroke impairment as either mild or severe upper limb impairment. Classifications for impairment severity have been proposed in prior literature based on a range of motor function scores (*Fugl-Meyer et al., 1975*; *Duncan et al., 1994*). The motor function domain is divided into the following: upper extremity (scored out of 36), hand (scored out of 10), wrist (scored out of 14), and coordination/speed (scored out of six) for a total of 66 indicating full performance of expected motor function for the upper limb (*Fugl-Meyer et al., 1975*).

## Participants

Complete demographics for participants with mild and severe impairments after stroke are detailed in Tables 1 and 2. Subjects that retained partial arm function and voluntary hand function, defined by an ability to grasp and release, were classified into the mild impairment group (UEFM score: 38–66, $n = 15$). Subjects that (1) could not complete the hand (10) and wrist (14) sections, (2) could not display at least one finger response to upper extremity reflex tests (/4), and (3) demonstrated an inability to actively extend the paretic wrist and fingers at least 20 degrees past neutral, were classified in the severe impairment group (UEFM score: 0–37, $n = 14$).

**Table 2 Severe impairment stroke survivor demographics.**

| Sub | M/F | Age (Years) | Time since stroke (Months) | Paretic arm | Dominant affected | Reach capacity (cm) | UEFM (max score = 66) |
|---|---|---|---|---|---|---|---|
| 1 | F | 69 | 12 | L | No | 13.8 | 10 |
| 2 | M | 57 | 120 | R | No | 30.2 | 24 |
| 3 | M | 56 | 16 | L | Yes | 14 | 8 |
| 4 | M | 63 | 11 | L | No | 27.8 | 29 |
| 5 | F | 68 | 112 | L | No | 43.4 | 23 |
| 6 | F | 44 | 30 | R | No | 22 | 25 |
| 7 | F | 69 | 9 | L | No | 27 | 14 |
| 8 | M | 51 | 5 | L | No | 19.2 | 10 |
| 9 | M | 54 | 43 | R | No | 4.1 | 7 |
| 10 | F | 70 | 401 | R | Yes | 4 | 14 |
| 11 | F | 78 | 8 | R | Yes | 5.3 | 12 |
| 12 | F | 63 | 25 | L | No | 14 | 22 |
| 13 | M | 71 | 28 | L | No | 7.3 | 13 |
| 14 | M | 57 | 49 | R | Yes | 4 | 16 |
| | 7M | 62.1 | 62.1 | 6R | 4Y | 16.9 | 16.2 |
| | 7F | $\pm 9.3$ | $\pm 104.2$ | 8L | 10N | $\pm 12.0$ | $\pm 7.1$ |

A two-sample t-test with equal variance and a one-tailed distribution was performed to evaluate differences between the mild and severe impairment groups. The mean age (Mild: $61.8 \pm 9.8$, Severe $62.14 \pm 9.3$ years) yielded a $p$-value of 0.46, suggesting an absence of statistically significant differences in age between the groups. Similarly, the mean time since stroke (Mild: $39 \pm 31$, Severe: $62 \pm 104.2$ months) yielded a $p$-value of 0.21, indicating no statistically significant differences. The assessment of the more affected or paretic limb between the groups (Mild: seven right arms, eight left, Severe: six right, eight left) and whether the paretic limb is also the dominant arm (Mild: nine yes, six no, Severe: four yes, 10 no) resulted in $p$-values of 0.42 and 0.09, respectively, suggesting no significant distinctions between the groups in these aspects.

However, distinctions emerged in metrics assessing functional outcomes. Notably, the maximum reaching ability (Mild: $35.58 \pm 8.9$, Severe: $16.86 \pm 12.0$ cm) yielded a significant $p$-value of $2.76E{-}05$, and the upper extremity Fugl-Meyer (UEFM) scores demonstrated marked disparity (Mild: $52.40 \pm 8.5$, Severe: $16.21 \pm 7.1$), accompanied by a significant $p$-value of $5.88E{-}13$. These statistical analyses collectively suggest that the two sample groups are not significantly differentiated in the metrics of age, time since stroke, and arm dominance. Metrics describing functional outcomes, however, such as maximum reaching ability and UEFM scores, are statistically significant indicators of distinction between the two sample groups.

## Experimental setup

Prior to the first data collection session, subjects were familiarized with the reaching task, and measurements of the chair height and distance of the chair from the table were
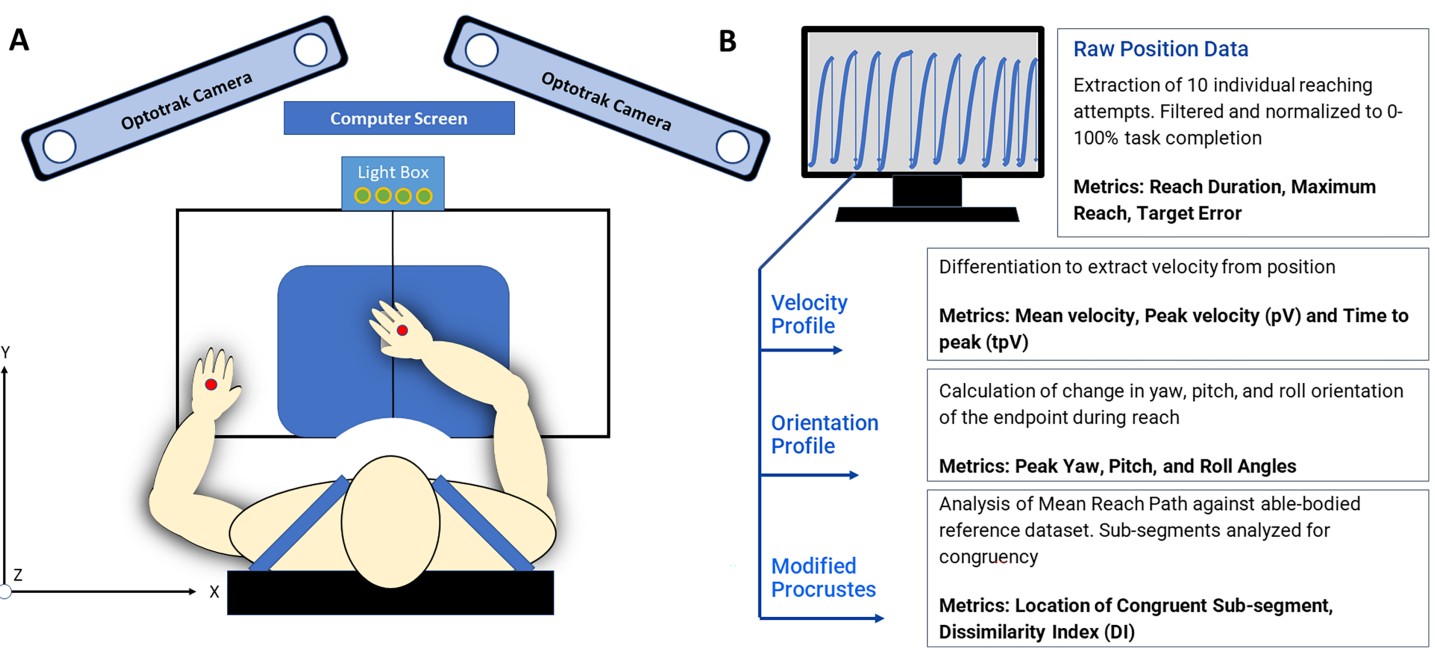

**Figure 1 (A) The reaching workspace and experimental protocol all data collection was conducted at the Mechanisms of Therapeutic Rehabilitation (MOTR) Lab at MedStar national rehabilitation hospital in Washington, DC. Markers placed on the hand dorsum are indicated in red. (B) 3D position data and kinematic analysis produced 3D positional data was evaluated with custom-written MATLAB scripts to extract individual curves and kinematic metrics such as movement variability, peak velocity, time to peak velocity, and target error.**

recorded. These measurements were adjusted to ensure the subject sat as close to the table as was comfortable and maintained a 90-degree resting angle at the elbow. The subject was fitted with trunk restraints to reduce appreciable trunk involvement in the forward-reaching movement (*Harrington et al., 2020*).

A single InfraRed Emitting Diode (IRED) optical marker was placed at the dorsal surface of each hand as appropriate given each subject's movement capability and resting hand position. A single target sensor was placed at 80% of the maximum reach of each individual subject. Placing the target within arm's reach rather than at maximum reach capacity ensured the subject would experience typical and moderate shoulder and elbow contribution and minimize uncomfortable or compensatory movements (*Ma et al., 2017*). The relative positions of the subject, two IRED markers, and the reaching workspace are depicted in Fig. 1.

Hand path kinematics were recorded using the Optotrak Certus motion capture system (Northern Digital Inc., Waterloo, Ontario, Canada) at a sampling frequency of 300 Hz, and the origin was calibrated at the front edge and center of the table at the beginning of each set of ten reaches. Optical tracking of upper extremity movement allows the collection of limb trajectory in terms of 3D Cartesian coordinates. Optotrak software was used to digitize the x-y plane in front of the subject and all movements were recorded with six degrees of freedom Optotrak cameras mounted surrounding and above the work-space.

Each subject completed a passive ideal hand path test in which the hand was passively moved to the target and back to represent movement without muscle activity. This

$$R = R_z\psi R_x\theta R_y\phi =$$

$$\begin{bmatrix} \cos\psi & -\sin\psi & 0 \\ \sin\psi & \cos\psi & 0 \\ 0 & 0 & 1 \end{bmatrix} \begin{bmatrix} \cos\theta & 0 & \sin\theta \\ 0 & 1 & 0 \\ -\sin\theta & 0 & \cos\theta \end{bmatrix} \begin{bmatrix} 1 & 0 & 0 \\ 0 & \cos\phi & -\sin\phi \\ 0 & \sin\phi & \cos\phi \end{bmatrix} =$$

$$\begin{bmatrix} \cos\psi\cos\theta & \sin\phi\sin\psi\cos\theta - \cos\phi\sin\theta & \sin\phi\sin\theta + \cos\phi\sin\psi\cos\theta \\ \cos\psi\sin\theta & \cos\phi\cos\theta + \sin\phi\sin\psi\sin\theta & \cos\phi\sin\psi\sin\theta - \sin\phi\cos\theta \\ -\sin\psi & \sin\phi\cos\psi & \cos\phi\cos\psi \end{bmatrix}$$

**Figure 2 Orientation of limb endpoint in 3-D space an intrinsic coordinate system centered at the hand was used to quantify movement refinement through the reach.**

measurement was used to verify and troubleshoot the collection of all positional data between the starting position and the target. Each subject completed two sets of the simple reaching test on two separate days with both the paretic and nonparetic arms. The forward-reaching task was initiated after a "Go" signal was indicated either in text on a screen or a light box placed within sight of the subject. Subjects were prompted with "When the 'Go' signal appears, quickly reach out to touch the target" to encourage rapid forward movement. Each testing session consisted of ten "Go" signals delivered at random intervals to ensure subjects did not anticipate movement initiation.

### Kinematic analysis

The discrete kinematic metrics of interest for this study are (1) peak velocity and time to peak velocity and (2) target accuracy. The continuous metrics of interest for this study are (1) endpoint orientation and (2) curve shape. The variables were chosen to represent movement strategy and performance, as they are often reported as related to the outcomes of therapy. The temporal location of the discrete kinematic landmarks during reach duration was used to characterize curve shapes highlighted by the Procrustes Analysis. The captured position data were transferred to MATLAB (The MathWorks Inc, Natick, MA, USA) software for analysis with custom-written scripts.

For the purposes of representing online movement refinement, we utilized the recommended method of a fixed local coordinate system with respect to the work-space (*Bai et al., 2014*; *Valevicius et al., 2018*; *Wu et al., 2005*). The y-axis extends directly forward and represents the primary distance covered during a reaching task. The x-axis extends laterally from the subject and the z-axis extends inferior to superior relative to the subject (Fig. 2).

The rotations of the distal coordinate system are described in terms of the proximal coordinate system. The first rotation was described as around the z-axis, the second was described around the x-axis, and the third was described around the y-axis, the longitudinal axis of the moving coordinate system. The rotation matrix in Fig. 2 describes the yaw-pitch-roll sequence of rotations; this was computed using consecutive data points for each incremental change in mean position during the forward reach. $\psi$ represents the yaw angle, $\theta$ represents the pitch angle, and $\phi$ represents the roll angle (*Sturm, Kerl &*

**Table 3 Kinematic metrics of hand movement included in data analysis.**

| Variables | Definition used for measurements |
|---|---|
| Reach duration | Time between a non-zero positive velocity followed by displacement in the positive y-direction, and a local displacement maxima which is immediately followed by a non-zero negative velocity. |
| | Normalized to 0–100% reach completion |
| Maximum reach | Farthest forward displacement achieved independently by subject when prompted to reach as far as they can while wearing trunk restraints |
| Mean velocity | The mean value during forward reach; derived from the mean velocity profile of all trials for an individual subject |
| Peak velocity | Maximum positive velocity achieved during reach duration and corresponding to the change from acceleration to deceleration |
| Time of peak velocity | Percentage of total reach duration where maximum peak velocity and change from acceleration to deceleration occurs |
| Yaw angle | $\psi$-Extrapolated by creating a rotation matrix A from position data every two consecutive time points, signifies the first rotation around the z-axis |
| Pitch angle | $\theta$-Extrapolated from rotation matrix A, rotation around the x-axis, |
| Roll angle | $\phi$-Extrapolated from rotation matrix A, represents the last rotation around the y-axis, *i.e.*, the longitudinal axis of the movement arm |
| Target error | Accuracy of the end displacement during individual reaches compared to a target placed at 80% max reach capacity |

*Cremers, 2015*). The definitions of these angles as well as other kinematic metrics of interest are listed in Table 3.

Velocity was extrapolated from the raw mean position data using the forward/backward/central differences in position data. Missing marker data were found for less than 10% of individual trials; missing data were corrected by extrapolating from adjacent position values. A low-pass fourth-order Butterworth filter with a cutoff frequency of 5 Hz was applied to the velocity data to reduce noise and distortion. The values of mean velocity, peak velocity, and the time-point where peak velocity was achieved were recorded for all subject data.

Finally, each trial of forward reaching was compared to the actual location of the target as recorded for each subject. The error tolerance was adjusted to account for reaches landing within the four squared inches of surface area of the target pad. Positive values of target error indicate when a subject stopped movement (identified by local maxima in displacement and subsequent movement in the negative y direction) before or at the target sensor. Negative values of target error represent when the subject has overshot or moved past the target.

## Modified procrustes analysis

Four reference curves were created from reach-to-target movements performed by two able-bodied volunteers. Able-bodied persons were recruited from the MedStar Rehabilitation Hospital volunteer population. Volunteers were asked to verify they had no diagnosis of a neurological or musculoskeletal disorder that could potentially influence movement control or reaching. In order to reduce the effects of hand dominance on the reference curves, one right-hand dominant and one left-hand dominant volunteer were selected. Three-dimensional position data was collected from both right and left limbs first

**Table 4 Kinematic metrics from literature and able-bodied reference curves.**

| Authors | Mean velocity (cm/s) | Peak velocity (cm/s) | Time to peak velocity (%) |
|---|---|---|---|
| *Murphy, Willén & Sunnerhagen (2011)* | – | 61.6 ± 9.4 | 46 ± 6.9 |
| *Patterson et al. (2011)* | – | 89 ± 13 (Comfortable) | 29.3 (Single exemplar) |
| | – | 121 ± 14 (Fast) | 32.8 (Single exemplar) |
| *van Dokkum et al. (2014)* | 42.6 ± 5 | 67.97 | – |
| Ref. Curve 1 | 21.24 | 57.62 | 18.20 |
| Ref. Curve 2 | 32.81 | 66.1 | 23.22 |
| Ref. Curve 3 | 23.58 | 39.72 | 26.75 |
| Ref. Curve 4 | 38.14 | 67.3 | 34.47 |

**Note:**
Reference curves 1 and 2 are left arm movements at a steady and rapid pace; curves 3 and 4 are right arm movements at a steady and rapid pace. Reference curves are used based on subject's paretic arm.

at a steady pace and then a rapid pace. The reference data set was used to create a mean healthy movement stereotype against which to analyze movements in the mild and severe impairment groups. The reference curves were compared against prior research to ensure curves were an appropriate representation of able-bodied movement. The values of the mean velocity, peak velocity, and time to peak velocity of our reference curves and values from other studies are compiled in Table 4. Reference curves were used only for the Procrustes trajectory analysis; each subject's velocity, target accuracy, and orientation variability were compared between the individual's paretic and non-paretic limbs.

Individual trials of reach-to target movements were extrapolated from raw kinematic data. The beginning of a movement was defined by displacement from the starting position and a non-zero positive velocity. The completion of a movement was defined by a local maxima in position immediately followed by a non-zero negative velocity. Reach detection was confirmed by visual inspection of each trial. Two sets of consecutive reaches were averaged to create a composite curve for each individual consisting of 20 trials. The reaching trajectory data was filtered by applying a low-pass fourth-order Butterworth filter with a cut-off frequency of 50 Hz to the trajectory data to account for minor variations in individual movement.

The reference trajectories were down-sampled to create ten fractions of the overall movement that were the same length as fractions of trajectories with motor impairments, in order to produce a dissimilarity profile of the overall reaching movement. Next, curve fragments composed of 35 time points across the reference and subject curves were compared. This required a novel modified Procrustes analysis that advanced point for point along the length of the subject and reference curves to identify segments that were congruent between both. In this particular application, the curves were not scaled, since the capacity to reach is specific to each subject. The index of dissimilarity, the sum of the squared Procrustes distance between each corresponding element in both curves, represents how incongruous the two segments may be, and was scaled to produce a value between 0 to 1, where 0 represents congruence between curves and 1 represents complete dissimilarity.
*Statistical analysis*

All statistical analyses were conducted in MATLAB with a statistical significance level of 0.05 (5%). When interpreting the results, $p$-values less than 0.05 were considered statistically significant and suggested the rejection of the null hypothesis.

Due to less than 50 subjects in either impairment group, an Anderson-Darling test for normalcy was performed on the kinematic metrics calculated from endpoint data (*Guthrie, 2020*). For the purposes of consistency in this article, all statistical analyses were performed using independent t-tests and N-way ANOVA. The one-way ANOVA is mathematically equivalent to an independent t-test when applied to only two groups (*Aron, Aron & Coups, 2019*). Kinematic metrics related to target error, peak velocity, and time point where peak velocity occurred were analyzed independently for differences due to impairment severity with a one-way ANOVA. Individual discrete kinematic measurements were compared in a two-way ANOVA against severity, whether the paretic limb is also the dominant limb, and which axis primarily contributed to the rotation. Separate two-way ANOVA was performed to analyze the results of the modified Procrustes analysis to interpret the significance of dissimilarity indices between mild and severe impairment groups. Kinematic measurements that appeared significantly different between the mild and severe impairment groups were then used to compute preliminary RSDI scores.

## RESULTS

All figures depict individual exemplars from the mild and severe impairment groups. The kinematic metrics for all subjects related to movement velocity, range of motion, and accuracy are reported in Tables A1 and A2. The Procrustean metrics for all subjects of translation, rotation/reflection, and scaling coefficients are reported in Tables A3 and A4.

### Kinematic findings

Figure 3 shows three example velocity profiles of subjects in the mild impairment group, and three example velocity profiles of subjects in the severe impairment group. Each black line indicates the averaged velocity profile across all reaching trials for a given subject, with standard deviation indicated by the blue shaded areas. The time when peak velocity is achieved is depicted in all profiles with a vertical red line.

In each mild impairment example shown, the peak velocity occurs towards the middle of the overall reach movement. For some subjects with severe impairment, the mean peak velocities were lower in magnitude (Mild: $0.97 \pm 0.40$ m/s, Severe: $0.76 \pm 1.42$ m/s) and occurred either closer to movement initiation or towards movement completion (Mild: $61.08\% \pm 17.08$, Severe: $49.55\% \pm 41.51$). The Anderson-Darling test for normalcy indicated for the scalar metrics of peak velocity (Mild: $p = 0.25$, Severe: $p = 0.73$) and velocity time location (Mild: $p = 0.30$, Severe: $p = 0.16$) that the hypothesis for normality was not rejected. The scalar metric of mean velocity (Mild: $p = 0.20$, Severe: $p = 0.95$) also showed evidence of normality.

Figures 4 and 5 show three example orientation profiles of subjects in the mild impairment group, and three example orientation profiles of subjects in the severe impairment group. Red curves indicate the averaged reach path of each subject, contrasted

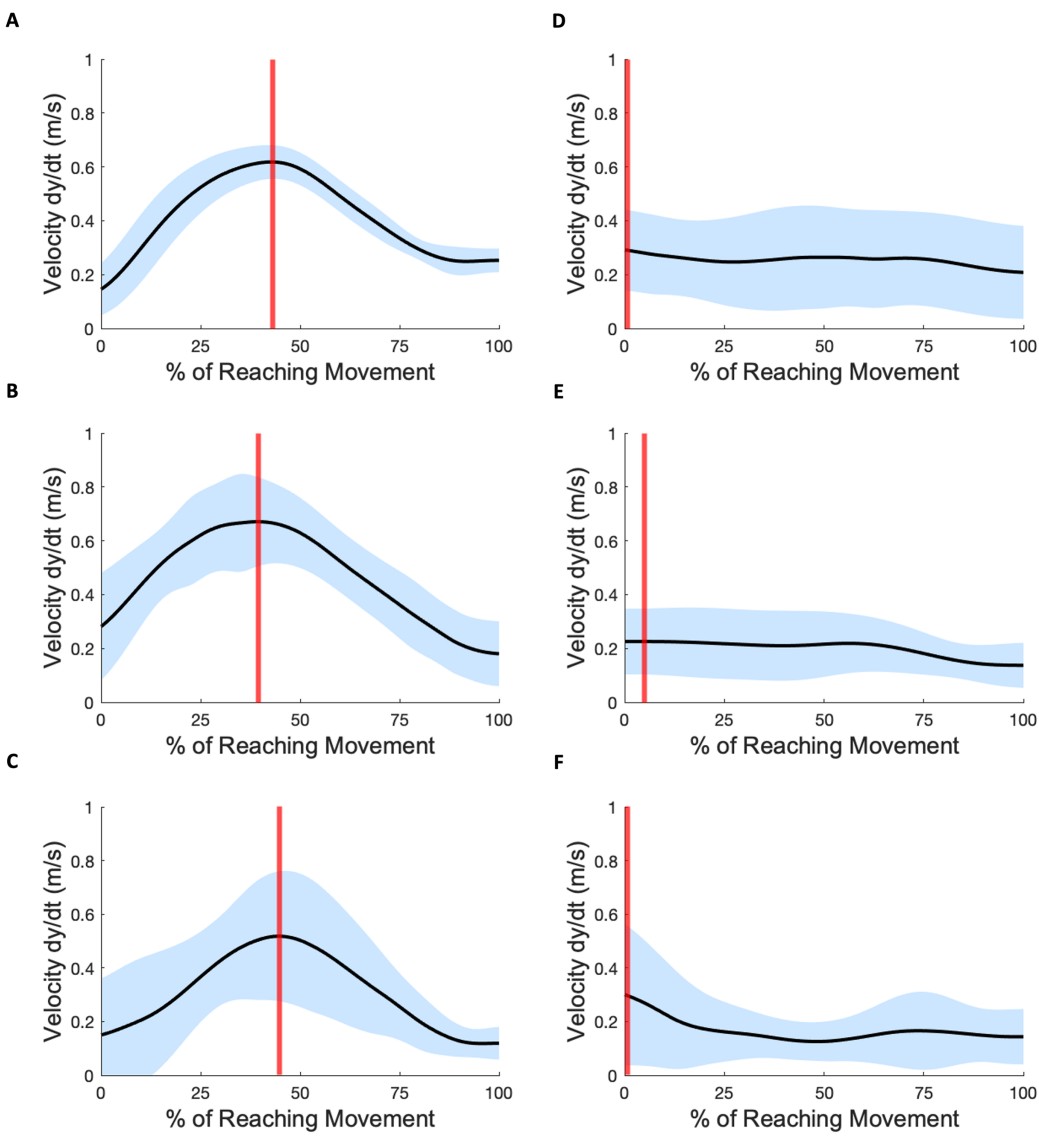

**Figure 3 Velocity profiles.** (A–C) Three subject exemplars with mild impairment. Black curves indicate average velocity across all subject reaching attempts; blue shading indicates standard deviation; vertical red lines indicate the time-location of peak velocity. (D–F) Three subject exemplars with severe impairment. The occurrence of peak velocity in these examples coincides with movement initiation.

with blue curves illustrating reference data sets. The three sub-graphs next to each subject's movement paths indicate the average yaw angle, average pitch angle, and average roll angle as calculated by rotation matrices during the movement. Subjects in the severe impairment group tended toward greater angular variability in the hand's orientation in the roll angle or about the y-axis as movement is completed (Mild peak roll angle: 111.22 ± 25.44, Severe: 86.34 ± 28.82). The reference reach curve shows some variability in orientation occurring about all three axes during movement initiation.

Finally, in order to assess subject accuracy, mean target error was calculated for each subject relative to the area of the target pad. All mean target percentage error values are

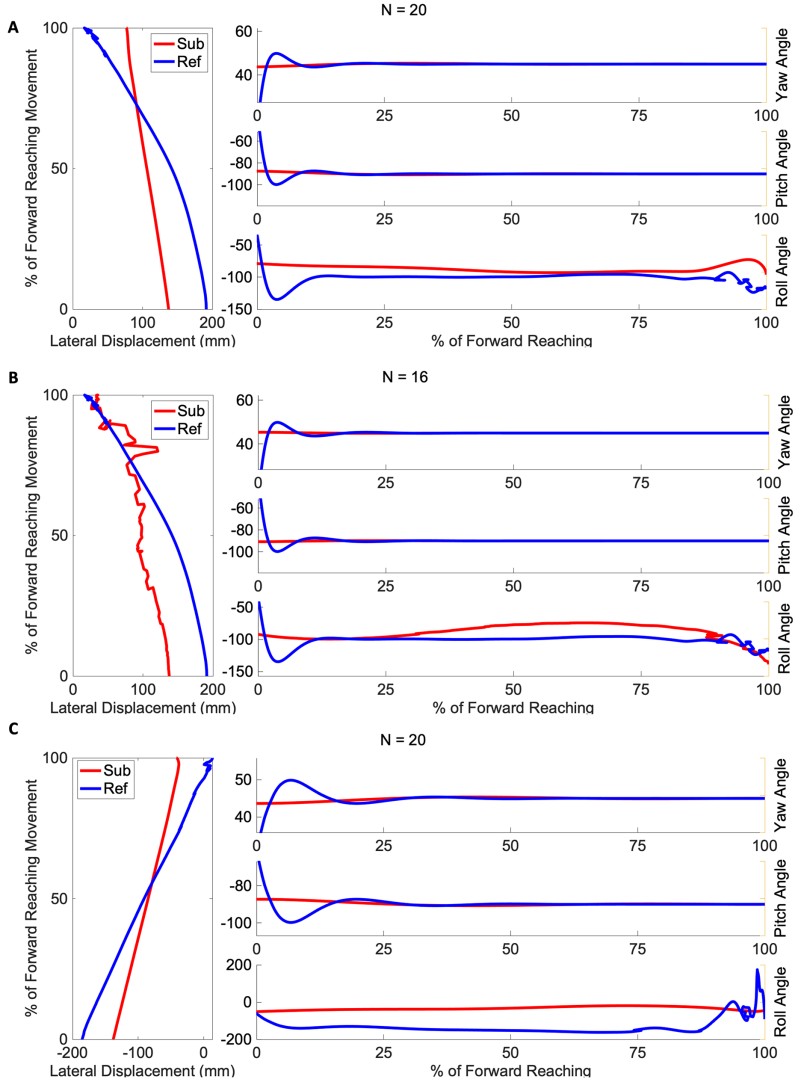

**Figure 4 Angular range of motion during forward movement from three representatives in the mild impairment group subject exemplars from the mild impairment group of average hand orientation.** (A and B) A right-handed reach path, (C) a left-handed reach path. Red lines indicate the average subject orientation profile, and blue lines indicate the reference orientation profile.

included in Tables A1 and A2. While both groups tended to undershoot the target rather than going beyond, the severe group had a larger mean target error (Mild: 9% ± 6, Severe: 43% ± 97) compared to the mild impairment group. The Anderson-Darling test indicated the hypothesis for normality was not rejected for the data collected on target error (Mild: $p = 0.69$ Severe: $p = 0.36$) was normally distributed.

Results of statistical analyses of kinematic metrics related to movement speed, range of motion and accuracy are summarized in Table 5. One-way ANOVA were performed to identify kinematic metrics that differed significantly between impairment groups.

The difference between the two groups' time to peak velocity, as shown by the red vertical lines in Fig. 3, was not statistically significant, indicated by a $p$-value of

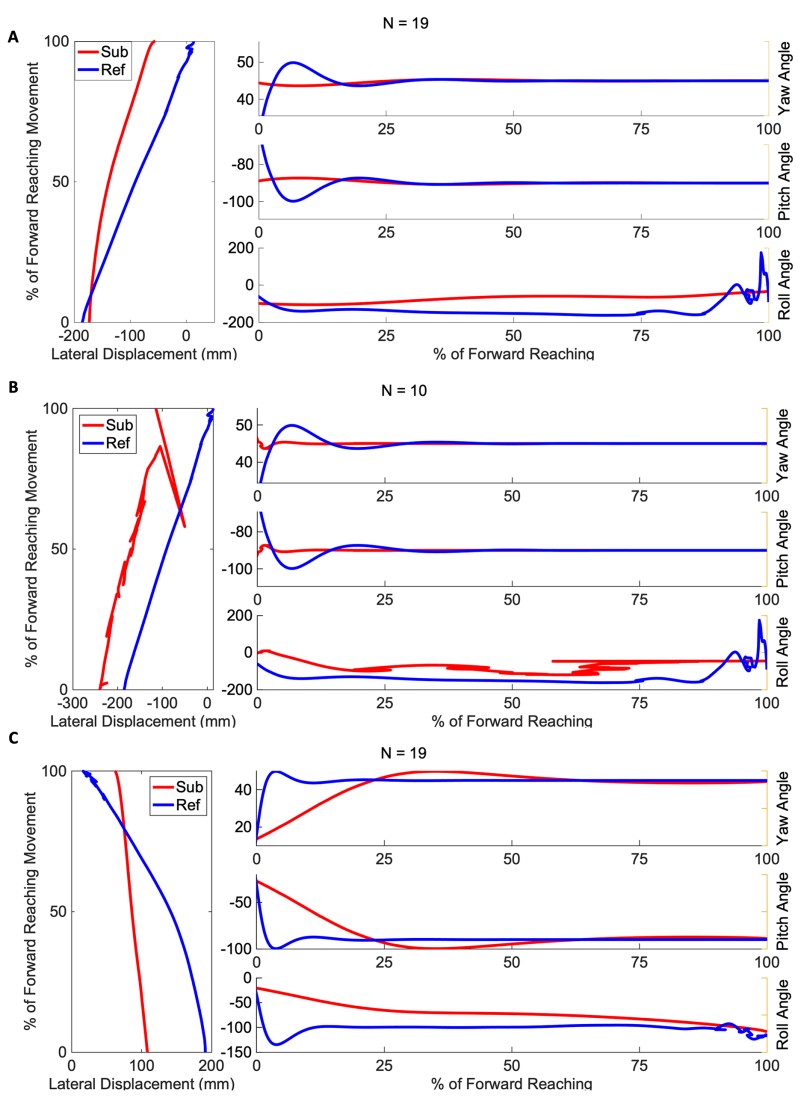

**Figure 5 Angular range of motion during forward movement from three representatives in the severe impairment group Subject exemplars from the severe impairment group of average hand orientation.** (A and B) A left-handed reach path, (C) a right-handed reach path. Red lines indicate average subject orientation profile, blue lines indicate reference orientation profile.

**Table 5 One-Way ANOVA of kinematic measurements between mild and severe groups.**

| Metric | Mild | Severe | *p*-value |
|---|---|---|---|
| Mean velocity (m/s) | 0.42 | 0.30 | 0.0173* |
| Peak velocity (m/s) | 0.97 | 0.76 | 0.5928 |
| Time to peak velocity (s) | 0.560 | 0.285 | 0.3307 |
| Time to peak velocity (%) | 61.08 | 49.55 | 0.2113 |
| Peak roll angle (deg) | 111.22 | 86.34 | 0.0202* |
| Target error | $-9 \pm 6$ | $-43 \pm 23$ | 0.0214* |

**Note:**
*P*-values were obtained from one-way ANOVA. Target error reported is percentage error. The *p*-values marked with an asterisk (*) are below the statistical significant level of 0.05, indicating significant differences between groups.

0.59. In contrast, the difference between the mean velocity of the impairment groups (Mild: 0.42 m/s, Severe: 0.30 m/s) was found to be significant with a *p*-value of 0.017. The peak roll angles, shown in the last sub-graph in each subject exemplar in Figs. 4 and 5, showed a statistically significant difference between the groups, with a *p*-value of 0.020. One-way ANOVA indicated a significant effect of group on the target error percentage by which subjects undershoot the target placed at 80% maximum reach capacity, with a *p*-value of 0.021.

In summary, the mild and severe groups differ significantly in the mean velocities, peak roll angle achieved during movement, and target error metrics. The kinematic findings of peak velocity and the time location during the movement when peak velocity is achieved were not found to be statistically significantly different between the two impairment groups.

## Modified procrustes analysis findings

A traditional Procrustes analysis was conducted to produce an overall dissimilarity index of each subject's mean reach path against reference data. When the complete subject reach path was compared to the complete reference reach path with a one-way ANOVA, there was no significant effect of severity on curve dissimilarity between the mild and severe groups (Mild dissimilarity index: 0.25, Severe: 0.20, $p = 0.62$).

Dissimilarity indices were calculated across ten equally sized segments of each subject's mean reach curve and compared with both the steady-paced and rapid-paced reference curves. The results are visualized as heatmaps of the median dissimilarity indices in each impairment group, shown in Fig. 6. Inspection of these heatmaps indicated higher dissimilarity indices as the movement ends in both mild and severe groups. Neither group appeared to be as dissimilar to the steady and rapid reference curves during the initiation and middle of overall movement.

When the groups were compared to the steady-paced reference, there was greater dissimilarity in the last three segments of movement (Mild median ending dissimilarity indices: (0.16, 0.6, 0.22), Severe: (0.18, 0.45, 0.2)). The rapid reference curve, which resulted from reference subjects being given the same prompt as the stroke subjects (*i.e.*, to move as quickly as they can), also resulted in greater dissimilarities (Mild median ending dissimilarity indices: (0.19, 0.46, 0.37), Severe: (0.15, 0.44, 0.55)) in the last three segments of movement.

The Procrustes method was then modified to compare segments, defined as 35 consecutive time-points, by advancing along the mean individual and reference reach curve point for point. In all cases of mild impairment, three of which are depicted in Fig. 7, the initial subject kinematic behavior appears most congruous to the initial control kinematic behavior. This is indicated by black lines connected each matched data pair between the red subject curves and the blue reference curves. These black lines are evident during movement initiation. Three subjects from the severe impairment groups are depicted in Fig. 8. These curves were not consistent in the location of the lowest dissimilarity indices, nor the time-duration during which movement progressed similarly to the steady and rapid reference curves.
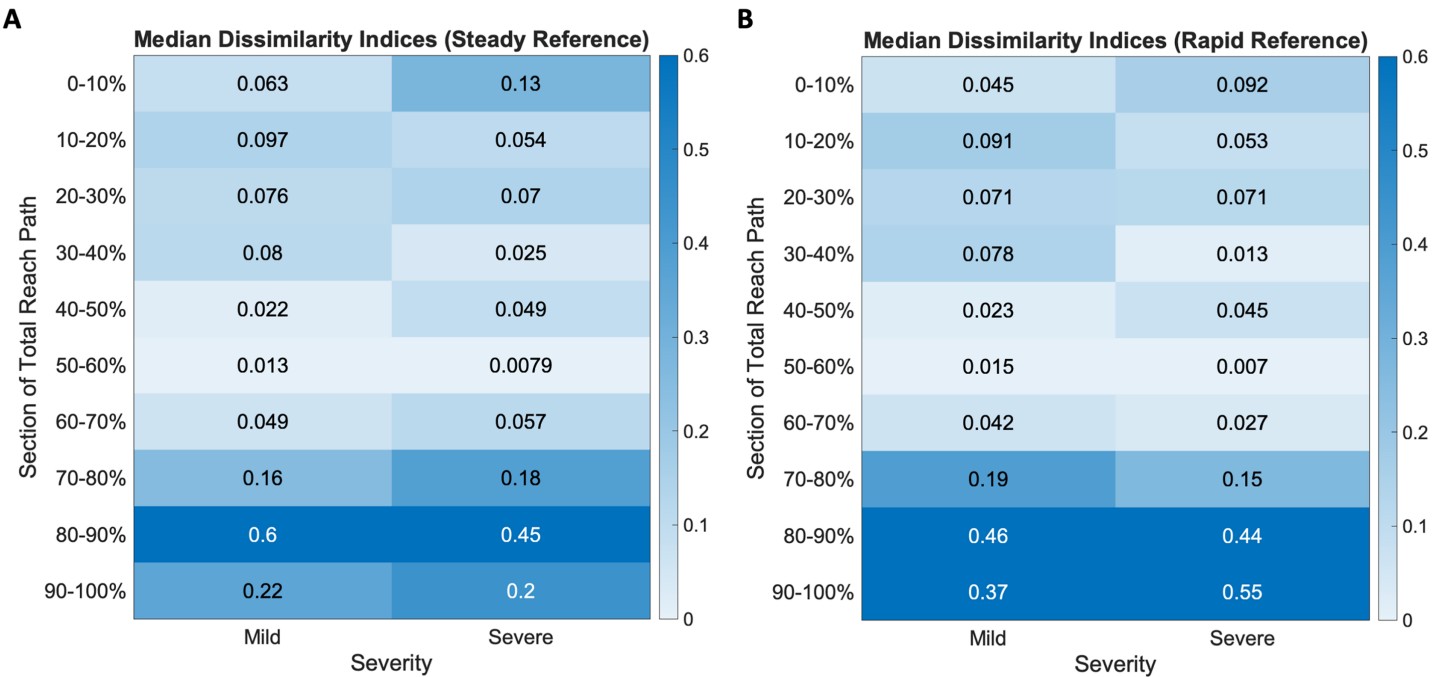

**Figure 6 Dissimilarity indices heatmaps show mild and severe impairment groups compared to the (A) steady and (B) rapid reference curves.** All groups show increased dissimilarity during movement completion, more so in the severe groups for both reference cases.

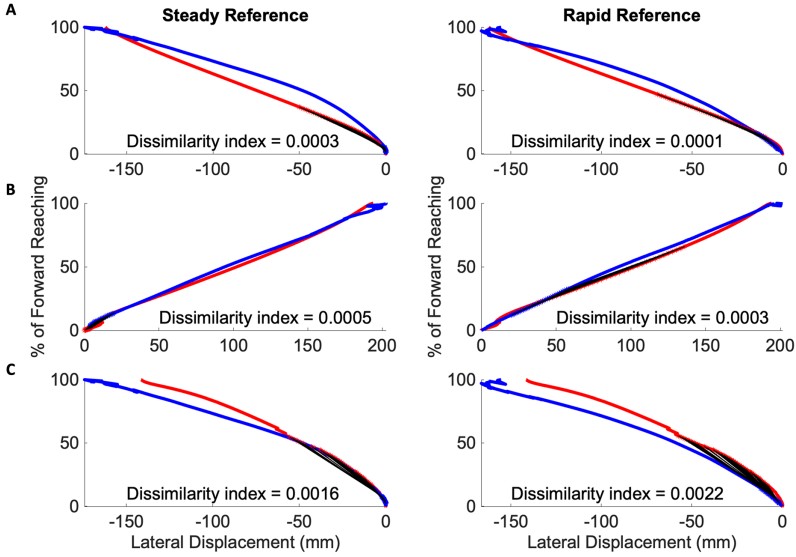

**Figure 7 Initiation movements in the mild impairment group remain congruous to healthy movement.** Modified Procrustes analysis of individuals with mild impairment shows movement initiation proceeds as in the steady (left in each pair of graphs) and rapid movement (right in each pair of graphs) curves. (A–C) Mild subject exemplars with black lines indicating matched data points between the red mean subject reach paths and blue reference reach paths.

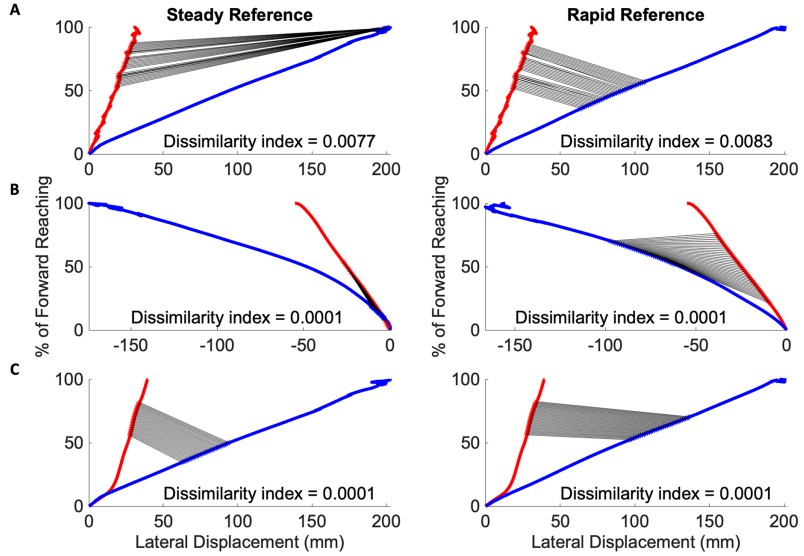

**Figure 8 Congruous movements in the severe impairment group modified Procrustes analysis of individuals with severe impairment against the steady (left in each pair of graphs) and rapid movement (right in each pair of graphs) curves.** (A–C) Severe subject exemplars with black lines indicating matched data points between the red mean subject reach paths and blue reference reach paths. Individuals with severe impairment do not have consistently located congruous behaviors to reference movement.

**Table 6 One-way ANOVA of procrustes transformation variables between mild and severe groups.**

| | Steady reference | | | Rapid reference | | |
|---|---|---|---|---|---|---|
| | **Mild** | **Severe** | ***p*-value** | **Mild** | **Severe** | ***p*-value** |
| Reference segment | | | 0.6288 | | | 0.0329* |
| Subject segment | | | 0.287 | | | 0.0285* |
| Rotation/reflection | | | 0.8813 | | | 0.2043 |
| Scaling | 3.62 | 2.49 | 0.3406 | 2.04 | 1.27 | 0.0397* |
| Translation | 51.48 | 240.72 | 0.1947 | 49.91 | 94.33 | 0.2889 |

**Note:**
The *p*-values marked with an asterisk (*) are below the statistical significant level of 0.05, indicating significant differences between groups.

Table 6 details the one-way analysis of variance in the rotation, scaling, and translation transformation variables found through the modified Procrustes analysis of the most congruent subject and reference segments. The mean scaling factors when compared to the smooth reference curve (Mild: 3.62, Severe: 2.49), and rapid reference curve (Mild: 2.04, Severe: 1.27) all indicate that the impairment groups demonstrated stretched movement, *i.e.*, the subjects took longer amounts of time than the reference to complete the specific segment of movement.

Table 7 shows the results of a two-way ANOVA, performed to analyze the influence of severity on the congruence of the subject curves to the steady and rapid reference curves. The only quantities found to differ significantly between impairment groups were location

**Table 7 Analysis of variance in severity and congruence to reference.**

| Source | Sum Sq | d.f. | p-value |
|---|---|---|---|
| Severity (Mild/Severe) | 31.1 | 1 | 0.4656 |
| Steady ref movement | 5,034.8 | 1 | 0* |
| Sub movement | 481 | 1 | 0.0096* |
| Severity * ref movement | 120.6 | 2 | 0.3619 |
| Severity * sub movement | 28.9 | 2 | 0.7747 |
| Ref * sub | 354 | 3 | 0.1378 |
| Severity (Mild/Severe) | 179.8 | 1 | 0.2059 |
| Rapid ref movement | 2,671.4 | 1 | 0.0001* |
| Sub movement | 2,826.8 | 1 | 0.0001* |
| Severity * ref movement | 227.8 | 2 | 0.3564 |
| Severity * sub movement | 182.8 | 2 | 0.4324 |
| Ref * Sub | 293.6 | 3 | 0.4415 |

Note:
Constrained (Type III) sum of squares. The p-values marked with an asterisk (*) are below the statistical significant level of 0.05, indicating significant differences between groups.

**Table 8 Analysis of variance in arm dominance and severity.**

| Source | Sum Sq | d.f. | p-value |
|---|---|---|---|
| Severity (Mild/Severe) | 1,732.7 | 1 | 0.0536* |
| Dominance of paretic | 3,277.3 | 1 | 0.0107* |
| Sub differences from steady ref | 5,711.4 | 2 | 0.0053* |
| Severity * dominance | 279.8 | 1 | 0.4186 |
| Severity * sub movement | 428.1 | 2 | 0.601 |
| Dominance * sub | 134.2 | 2 | 0.85 |
| Severity (Mild/Severe) | 36.1 | 1 | 0.7815 |
| Dominance of paretic | 82.6 | 1 | 0.6753 |
| Sub differences from rapid ref | 14,635.2 | 2 | 0.0001* |
| Severity * dominance | 1,188.4 | 1 | 0.1231 |
| Severity * sub movement | 981.5 | 2 | 0.3611 |
| Dominance * sub | 709.3 | 2 | 0.4738 |

Note:
Constrained (Type III) sum of squares. The p-values marked with an asterisk (*) are below the statistical significant level of 0.05, indicating significant differences between groups.

of the congruent subject segment ($p = 0$) and the steady reference segment to which it coordinated ($p = 0.01$).

Table 8 shows the results of a two-way ANOVA, performed to analyze the influence of severity and arm dominance on the presence of congruent segments in the subject curves. In the case of the steady curve, severity was found to influence significant differences ($p = 0.05$), as with arm dominance ($p = 0.011$), on whether segment congruence was found. In the case of the rapid reference curve, severity and arm dominance did not contribute to significant differences between the impairment groups.

**Table 9 Analysis of variance in influence of arm dominance and severity on size of congruent segments.**

| Source | Sum Sq | d.f. | p-value |
|---|---|---|---|
| Severity (Mild/Severe) | 1,072.27 | 1 | 0.0342* |
| Dominance of paretic | 393.52 | 1 | 0.1828 |
| Subject movement | 965.76 | 2 | 0.1229 |
| Severity * Dominance | 1,042.78 | 1 | 0.0364* |
| Severity * Sub movement | 180.53 | 2 | 0.6514 |
| Dominance * Sub | 56.9 | 2 | 0.8718 |

Note:
Constrained (Type III) sum of squares. The p-values marked with an asterisk (*) are below the statistical significant level of 0.05, indicating significant differences between groups.

Table 9 shows a two-way ANOVA, performed to analyze the influence of severity and dominance on the time-length of subject segments that appeared most congruent to reference curves. The impairment severity classification of the subject had a significant main effect on the length of congruence of the subject segment, p-value of 0.034. Where there were no significance of the other main effects, the two-way interaction of severity and arm dominance had a p-value of 0.036.

In summary, when analyzed against the rapid reference curve, the subject groups differed significantly in scaling coefficients, and the location within the overall movement of congruent sub-movements between subject and reference curves. As seen in Table 7, impairment groups differ significantly in the subject movement that most replicates a phase of movement in the steady and rapid reference curves. The effects of severity on the ability to consistently perform steady reaching movements were significantly mediated by arm dominance, seen in Table 8. As indicated in Table 9, the population marginal means of the subjects with mild impairment with a paretic non-dominant limb, and severe impairment with a paretic non-dominant limb are significantly different.

The population marginal means for both groups of impairment where the paretic limb is also the dominant limb did not have any significant differences. Other metrics derived from the traditional and modified Procrustes analysis that did not indicate significant difference between impairment groups included whole curve dissimilarity, rotation/reflection matrices, and translation coefficients.

## Preliminary severity and dissimilarity scores

Metrics related to kinematics and the modified Procrustes analysis that showed significant differences between the mild and severe populations were used to compute RSDI-severity and RSDI-dissimilarity sub-scores. The severity sub-score comprised of velocity, orientation, and accuracy elements while the dissimilarity sub-score comprised of dissimilarity indices of the ending movements, and the location and length of the reference segment that was found to be most congruous. The scaling components were also included in the computation of the dissimilarity sub-score.

The preliminary RSDI sub-scores computed using these metrics were classified in terms of likely rehabilitation goals. Subjects with a higher severity indices and lower dissimilarity

**Table 10 Cross tabulation table for upper extremity Fugl-Meyer (UEFM) and reach severity & dissimilarity index (RSDI).**

|  | RSDI | | |
|  | Speed emphasis | Strength emphasis | Combined emphasis |
| --- | --- | --- | --- |
| Mild | 0 | 10 | 5 |
| Severe | 5 | 7 | 2 |

indices due to low mean velocities, low peak angular values, and high target error, may benefit from a classification that prioritizes speed-focused goals. Such subjects were given a Speed Emphasis classification. Alternatively, subjects with lower severity indices and higher dissimilarity indices were scored as such due to high dissimilarity to the reference movement, or elongated movement behaviors, implying a need for Strength Emphasis to produce stable movements. Subjects with comparable severity and dissimilarity indices were classified as Combined Emphasis. These classifications compared with the UEFM mild and severe classifications are cross tabulated in Table 10.

The first row in Table 10 shows that of the 15 subjects classified in the mild impairment group according to the UEFM test, 10 received a Strength Emphasis and five received a Combined Emphasis. This is consistent with the clinical observation that persons with mild impairment continue to be able to reach forward quickly while compensating for muscle weakness and loss of agility. The second row indicates that of the 14 subjects classified in the severe impairment group by the UEFM test, five can be reclassified as Speed Emphasis, seven as Strength Emphasis, and two as Combined Emphasis.

## DISCUSSION

In this article, we considered metrics of motor performance that can be derived through three-dimensional endpoint kinematic analysis. Additionally, we utilized a novel modified Procrustes Shape Analysis methodology to identify subtrajectories during upper extremity motor performance that exhibit similarities to reference movement data. Those kinematic and similarity findings that were found to indicate significant differences between subjects in mild and severe impairment groups were then used to compute preliminary Reach Severity and Dissimilarity Indices, a proposed summary score that may quantify motor performance and recovery, and inform rehabilitative focii for individuals after stroke.

Impaired upper extremity function is a common sequalae after injuries to the neural apparatus. When such injuries occur as a result of hemorrhagic strokes, many activities of daily living (ADLs) are affected, contributing to functional limitations and disability (*Cirstea & Levin, 2000*; *Oosterwijk et al., 2018*). The measurement of human motor performance after stroke can serve multiple purposes, particularly in individualizing rehabilitation treatments, informing clinical decision-making, evaluating efficacy, and distinguishing improvements in impairment severity (*de los Reyes-Guzmán et al., 2014*). In particular, collecting movement trajectories allows the evaluation of how specific movements emerge and how they are spatially and temporally segmented as the movement progresses (*Passos et al., 2023*).

The relationship between performance and dissimilarity in trajectory is heavily dependent on the context. In this article, reaching was selected as a functional sub-movement of many ADLs and as a representative sample of the upper extremity's motor abundance. While existing approaches in trajectory analysis have primarily focused on enhancing the precision of evaluation dissimilarity (*Passos et al., 2023*), we describe the modification of Procrustes methodologies to locate subtrajectories that exhibit the greatest similarity to a reference dataset, to identify phases of movement that can be used to qualify an individual's impairment severity. Human movement variability can be due to task complexity, restrictions of the environment, or ongoing error corrections during the movement (*Bernstein, 1967*); therefore subtrajectories that remain similar between subject and reference movement are of greater interest than whole trajectories in creating a quantitative and objective measure of motor performance (*Buchin et al., 2011*).

Motion analysis datasets are often large with a wide range of values and information. Creating a clinical outcome measure, or subject-specific summary index of motor performance is an increasingly popular method of synthesizing large amounts of clinical data into a single objective score (*Cutti, Parel & Kotanxis, 2017*). Current methods include the Arm Profile Score (APS) (*Ueyama, 2021*), a score based on kinematic data which uses differences between subject and reference datasets. The overall severity of movement can be summarized from 13 different scores representing different joint angles of the arm. Another measure, the Pediatric Upper Limb Motion Index (PULMI), is also currently used to quantify upper extremity movement pathology in children with unilateral cerebral palsy. The PULMI alternatively is a single composite score (*Mailleux et al., 2017*). Both these scores are inspired by the Gait Profile Score (*Baker et al., 2009*), which sets the precedent for measuring root-mean-square differences between subject kinematic data and reference datasets of typical motor performance. Therefore, this article's methodology of identifying differences between subject and reference data for the purposes of creating an objective quantitative score of arm severity and dissimilarity is founded on methods well-established in the disciplines of rehabilitation and clinical movement analysis.

We used endpoint kinematics describing movement qualities such as speed, accuracy, and functional range of motion (ROM) to complement dissimilarity scores as these have been validated in prior literature as significant representations of motor performance (*Wagner, Rhodes & Patten, 2008*; *Murphy, Willén & Sunnerhagen, 2011*; *Jaspers et al., 2011a*; *Merlo et al., 2013*). The subjects classified in the mild impairment group in this study achieved higher mean velocities (0.97 m/s) than their counterparts (0.76 m/s) in the severe impairment group. Higher target error may be correlated with diminished ability to sub-correct movements during the final phase of movement where precision and accuracy is prioritized. Earlier motor control decision-making prioritizes speed and minimization principles. The data thus lends some support to the observation that response time and target accuracy are disrupted after stroke but not physical capability of initiating movement. We also noted that the mild group individuals have higher peak roll angles, with a mean value of 111.22 as opposed to a mean value of 86.34 degrees for the severe group, indicating individuals with severe impairments performed less rotation around the y-axis. This may suggest the presence of maladaptive joint movements, potentially due to a

compensatory adjustments between the elbow and shoulder. The findings have clinical implications, suggesting possible rehabilitative strategies to promote more effective practice by placing targets in different positions in front of the subject.

Some limitations of the proposed methodology include the inability of the optical markers used to measure more complex tasks such as those required in self-care and hygiene (*Valevicius et al., 2018*). While reaching is a good representation of the ability of the upper extremity to complete movements where point-to-point reaching is a sub-movement, it is beyond the scope of this study to identify further optimal categories of movements. Therefore, the preliminary severity and dissimilarity subscores presented in this article can only identify rehabilitative targets regarding an individual's reaching ability. The relatively low cost application of this methodology through MATLAB can be modified for further movements, given that a reference dataset is first constructed using movements completed by healthy volunteers. The markers utilized can also be replaced in future studies with acceleration and position markers that can be placed on facets of the body not particularly accessible to optical tracking systems.

This method is limited in creating a thoroughly subject-specific model in that it cannot identify pathological involvement of a particular limb segment or joint. Individualizing all functional measures is very difficult (*Bolsterlee, Veeger & Chadwick, 2013*); however, by incorporating additional markers on all segments of the upper extremity provides an interesting expansion of the methodology described here. We presented a preliminary work of inter-joint coordination being represented and evaluated with the modified Procrustes methodology in July 2023 at the Engineering in Medicine and Biology Conference (*Zaidi & Harris-Love, 2023*). In this study, the trajectories of the upper arm, forearm, and hand segments were evaluated for similar subtrajectories as a potential method for identifying time-locations of inter-joint coordination during the reach-to-target task.

Another limitation of the current methodology is the inability to differentiate which aspect of functional limitations leads to a specific subtrajectory appearing to be similar to reference movement. During forward reaching, motor control is characterized by the need for movement initiation, target precision, and error subcorrections. Errors may be generated through motor performance prediction error, where the ensuing movement does not match the expected movement (*Ranjan & Smith, 2020*), or sensory prediction error, where sensory feedback informs how the ensuing movement must be corrected. In the case of the mild impairment group, the dissimilarity indices of specific events within the reaching task are of particular interest, and imply that some movement behavior is preserved that is disrupted with severe impairment. A most interesting finding of the modified Procrustes analysis is that severity has a significant interaction effect, along with hand dominance, on whether a subject replicates reference behavior while initiating reach or at some point during the reach task. The relative timing of the peak velocity within the first phase of movement follows prior literature describing the initiation of movement being based on anticipation of the task and not sensory feedback. Currently it is outside the scope of this study to identify why the initiation behaviors appear similar between subject and reference data. We are also not able to identify underlying strategies that lead to some

similar trajectories in individual cases from the severe impairment group, whether due to some retained ability to incorporate sensory feedback or internally generate motor performance error. Furthering this methodology with behavioral analysis that lends insight to underlying motor functionality would be a very interesting and significant contribution towards creating an effective summary index of severity.

Although three-dimensional motion analysis is a valuable tool for evaluating the upper extremity, its application is complicated by the absence of periodic movements and the presence of numerous degrees of freedom. It is also difficult to translate to the clinical setting with repetitious measurements required to monitor efficacy of treatment, true motor recovery, and identify new individual targets for rehabilitation. The clinical interpretation of measured movement pathology through three-dimensional motion analysis can be made simpler by utilizing summary scores such as the proposed Reach Severity and Dissimilarity Index. In the clinical setting, a subject demonstrating congruous movement initiation may be instructed to focus on precision exercises and visual feedback incorporation, while a subject demonstrating congruous movement completion may practice speed exercises and need not emphasize target accuracy. Therefore, we anticipate the methodology described here may translate well to the clinical setting without requiring changes to data collection modalities, extensive and subjective evaluations, or increasing patient burden.

## CONCLUSIONS

While rehabilitation efforts can be effectively informed by clinical observation in the case of individuals with mild functional impairments, individuals exhibiting severe impairments require a deeper investigation of when and how deficits emerge. Though the upper extremity is neither cyclical nor stereotyped in its movement like the lower extremity, nevertheless measurements of gait deviation can guide analogous measures of severity and dissimilarity for the arm during functional sub-movements such as the reach and grasp cycle.

The modified Procrustes method produced intriguing results that are supported by clinical observations; namely that mild impairment does not necessarily lead to a disruption in the ability to initiate rapid movement. By comparing curved paths point by point, clinicians may pinpoint when a disruption in movement occurs. This creates the possibility for movement tracking to remain simple yet effective, so that it can be incorporated into the clinical setting without increasing patient burden.

The upper extremity presents a rich platform for studying the motor system and how it is affected by the physical world around it and the internal world that controls and communicates through it. Through advancing the kinematic questions explored in this study and understanding the specific control parameters and factors that constrain and alter function, we hope that the impairment and functional limitations correlated with stroke may be minimized and thus prevented from translating to disability in social functioning. By creating a comprehensive and objective clinical tool to select rehabilitative strategies that can serve each individual's specific needs, we anticipate the impact of stroke on disability and quality of life may be appreciably reduced.

# APPENDIX

**Table A1 Kinematic findings for mild impairment group, $n = 15$.**

| Sub | Velocity profile | | | | Orientation and TE | | | |
|---|---|---|---|---|---|---|---|---|
| | $V_{mean}$ (m/s) | pV (m/s) | tpV (%) | tpV (s) | $\psi_{max}$ | $\theta_{max}$ | $\phi_{max}$ | Target error (%) |
| 1 | 0.33 | 0.90 | 69.43 | 0.810 | 49.89 | 99.78 | 96.31 | 14 ± 2 |
| 2 | 0.58 | 1.56 | 93.60 | 0.780 | 49.89 | 99.78 | 138.88 | −24 ± 6 |
| 3 | 0.42 | 0.62 | 43.20 | 0.360 | 49.89 | 99.78 | 151.24 | −3 ± 3 |
| 4 | 0.47 | 0.67 | 39.50 | 0.263 | 49.89 | 99.78 | 117.46 | −10 ± 5 |
| 5 | 0.48 | 1.29 | 67.33 | 0.673 | 49.89 | 99.78 | 112.49 | −18 ± 3 |
| 6 | 0.44 | 1.20 | 73.60 | 0.613 | 49.89 | 99.78 | 117.59 | −2 ± 5 |
| 7 | 0.44 | 0.65 | 48.00 | 0.320 | 49.89 | 99.78 | 68.65 | 9 ± 4 |
| 8 | 0.38 | 0.79 | 59.60 | 0.497 | 49.89 | 99.78 | 113.78 | −9 ± 8 |
| 9 | 0.44 | 1.87 | 84.86 | 0.990 | 49.89 | 99.78 | 151.28 | −13 ± 4 |
| 10 | 0.31 | 0.52 | 44.80 | 0.373 | 49.89 | 99.78 | 96.13 | −12 ± 10 |
| 11 | 0.49 | 0.85 | 54.00 | 0.630 | 49.89 | 99.78 | 122.43 | −11 ± 8 |
| 12 | 0.42 | 0.97 | 67.60 | 0.563 | 49.89 | 99.78 | 100.69 | −12 ± 6 |
| 13 | 0.46 | 1.25 | 70.80 | 0.590 | 49.89 | 99.78 | 109.08 | −23 ± 6 |
| 14 | 0.22 | 0.50 | 33.60 | 0.280 | 49.89 | 99.78 | 61.26 | −4 ± 10 |
| 15 | 0.41 | 0.83 | 66.33 | 0.663 | 49.89 | 99.78 | 111.00 | −10 ± 4 |
| Mean | 0.42 | 0.97 | 61.08 | 0.560 | 49.89 | 99.78 | 111.22 | −9 |
| STD | ± 0.085 | ± 0.40 | ± 17.08 | ± 0.21 | | | ± 25.44 | ± 6 |

**Note:**
$V_{mean}$, mean velocity; pV, peak velocity, tpV, time to peak velocity; $\psi$, maximum yaw; $\theta$, pitch; $\phi$, roll angles; TE, target error, negative value indicates completing short of the target.

**Table A2 Kinematic findings for severe impairment group, $n = 14$.**

| Sub | Velocity profile | | | | Orientation and TE | | | |
|---|---|---|---|---|---|---|---|---|
| | $V_{mean}$ (m/s) | pV (m/s) | tpV (%) | tpV (s) | $\psi_{max}$ | $\theta_{max}$ | $\phi_{max}$ | Target error (%) |
| 1 | 0.25 | 0.29 | 0.83 | 0.003 | 49.89 | 99.78 | 36.15 | 2 ± 18 |
| 2 | 0.16 | 0.20 | 0.50 | 0.003 | 49.89 | 99.78 | 74.82 | −15 ± 3 |
| 3 | 0.20 | 0.23 | 5.00 | 0.020 | 49.89 | 99.78 | 50.99 | −20 ± 12 |
| 4 | 0.62 | 5.63 | 100.00 | 1.000 | 49.89 | 99.78 | 120.14 | −90 ± 23 |
| 5 | 0.48 | 0.76 | 63.50 | 0.423 | 49.89 | 99.78 | 105.69 | 36 ± 7 |
| 6 | 0.26 | 0.35 | 100.00 | 0.567 | 49.89 | 99.78 | 76.39 | −32 ± 11 |
| 7 | 0.41 | 0.73 | 71.76 | 0.407 | 49.89 | 99.78 | 120.90 | −9 ± 9 |
| 8 | 0.41 | 0.49 | 100.00 | 0.500 | 49.89 | 99.78 | 115.16 | −44 ± 10 |
| 9 | 0.33 | 0.40 | 42.50 | 0.113 | 49.89 | 99.78 | 99.48 | −33 ± 44 |
| 10 | 0.32 | 0.42 | 92.00 | 0.307 | 49.89 | 99.78 | 109.38 | −20 ± 27 |
| 11 | 0.05 | 0.06 | 2.00 | 0.003 | 49.89 | 99.78 | 55.17 | −19 ± 21 |
| 12 | 0.44 | 0.65 | 76.00 | 0.507 | 49.89 | 99.78 | 112.56 | −71 ± 18 |

| | Velocity profile | | | | Orientation and TE | | | |
|---|---|---|---|---|---|---|---|---|
| Sub | $V_{mean}$ (m/s) | pV (m/s) | tpV (%) | tpV (s) | $\psi_{max}$ | $\theta_{max}$ | $\phi_{max}$ | Target error (%) |
| 13 | 0.17 | 0.30 | 0.67 | 0.003 | 49.89 | 99.78 | 60.28 | $-117 \pm 23$ |
| 14 | 0.09 | 0.11 | 39.00 | 0.130 | 49.89 | 99.78 | 71.70 | $-178 \pm 97$ |
| Mean | 0.30 | 0.76 | 49.55 | 0.285 | 49.89 | 99.78 | 86.34 | $-43$ |
| STD | $\pm0.16$ | $\pm1.42$ | $\pm41.51$ | $\pm0.30$ | | | $\pm28.82$ | $\pm97$ |

Note:
$V_{mean}$, mean velocity; pV, peak velocity, tpV, time to peak velocity; $\psi$, maximum yaw; $\theta$, pitch; $\phi$, roll angles; TE, target error, negative value indicates completing short of the target.

**Table A3 Procrustean measures of dissimilarity for mild impairment group.**

| | Steady reference curve | | | | Rapid reference curve | | | |
|---|---|---|---|---|---|---|---|---|
| Subject | Location (R/S) (%) | det (T) | b | c | Loc (%) | det (T) | b | c |
| 1 | 30 == 69 | 1 | 2.56 | 70.92 | 32 == 68 | 1 | 1.38 | 34.83 |
| 2 | 3 == 51 | 1 | 11.05 | 5.39 | 6 == 56 | −1 | 3.93 | 12.41 |
| 3 | 21 == 54 | −1 | 1.19 | 38.20 | 15 == 30 | −1 | 0.94 | 17.06 |
| 4 | 10 == 21 | 1 | 1.54 | 3.03 | 59 == 54 | 1 | 1.79 | 193.74 |
| 5 | 29 == 66 | 1 | 3.79 | 100.86 | 63 == 17 | −1 | 2.63 | 10.43 |
| 6 | 7 == 63 | 1 | 11.21 | 7.05 | 66 == 12 | 1 | 3.72 | 15.00 |
| 7 | 19 == 49 | −1 | 1.53 | 19.68 | 62 == 57 | −1 | 2.02 | 255.52 |
| 8 | 8 == 46 | −1 | 1.63 | 6.61 | 54 == 15 | −1 | 1.21 | 12.60 |
| 9 | 8 == 16 | 1 | 0.22 | 0.81 | 83 == 16 | −1 | 2.46 | 21.70 |
| 10 | 50 == 45 | 1 | 1.56 | 111.87 | 49 == 53 | 1 | 0.94 | 51.20 |
| 11 | 57 == 46 | −1 | 1.99 | 203.81 | 39 == 19 | 1 | 1.49 | 7.08 |
| 12 | 31 == 65 | −1 | 3.12 | 103.69 | 61 == 19 | 1 | 2.11 | 6.59 |
| 13 | 10 == 63 | −1 | 9.33 | 6.46 | 63 == 10 | 1 | 4.02 | 11.50 |
| 14 | 66 == 45 | 1 | 0.54 | 26.66 | 62 == 14 | 1 | 0.34 | 75.37 |
| 15 | 30 == 67 | 1 | 3.02 | 67.22 | 67 == 33 | 1 | 1.62 | 23.56 |
| Mean | | | 3.62 | 51.48 | | | 2.04 | 49.91 |
| STD | | | 3.72 | 58.25 | | | 1.13 | 74.24 |

Note:
Location indicates the time-step in reference (R) and subject (S) curves that are most congruent. T, b, and c are Procrustean values that quantify the differences between these segments of the reference and subject curves. T indicates the Rotation/Reflection component of conforming the subject curve to the reference curve (a determinant of 1 indicates a rotation and −1 indicates a reflection), b indicates the scaling coefficient, and c indicates the magnitude of the translation vector.

**Table A4 Procrustean measures of dissimilarity for severe impairment group.**

| | Steady reference curve | | | | Rapid reference curve | | | |
|---|---|---|---|---|---|---|---|---|
| Subject | Location (R/S) (%) | det(T) | b | c | Loc (%) | det(T) | b | c |
| 1 | 12 == 60 | 1 | 1.19 | 23.85 | 13 == 58 | −1 | 0.97 | 27.86 |
| 2 | 57 == 24 | 1 | 1.38 | 160.57 | 20 == 58 | 1 | 0.75 | 57.74 |
| 3 | 92 == 66 | −1 | 9.37 | 2029.10 | 28 == 65 | 1 | 0.78 | 17.75 |
| 4 | 10 == 38 | 1 | 0.24 | 5.83 | 12 == 38 | 1 | 0.19 | 6.56 |

(Continued)

| Subject | Steady reference curve | | | | Rapid reference curve | | | |
|---|---|---|---|---|---|---|---|---|
| | Location (R/S) (%) | det(T) | b | c | Loc (%) | det(T) | b | c |
| 5 | 13 == 60 | 1 | 1.76 | 13.00 | 15 == 61 | −1 | 1.43 | 25.10 |
| 6 | 11 == 57 | −1 | 2.86 | 28.01 | 55 == 8 | 1 | 1.58 | 28.20 |
| 7 | 13 == 71 | 1 | 1.84 | 11.00 | 15 == 72 | −1 | 1.5 | 19.05 |
| 8 | 20 == 39 | −1 | 1.18 | 36.28 | 27 == 22 | −1 | 0.88 | 21.37 |
| 9 | 17 == 26 | −1 | 3.83 | 43.65 | 43 == 43 | −1 | 1.58 | 112.37 |
| 10 | 59 == 19 | 1 | 1.80 | 245.35 | 19 == 53 | −1 | 1.00 | 120.51 |
| 11 | 64 == 37 | −1 | 4.71 | 716.18 | 37 == 52 | −1 | 2.33 | 274.91 |
| 12 | 5 == 15 | 1 | 1.12 | 0.54 | 83 == 67 | −1 | 3.01 | 512.38 |
| 13 | 22 == 63 | 1 | 0.77 | 34.93 | 64 == 36 | 1 | 0.61 | 31.34 |
| 14 | 18 == 42 | −1 | 2.83 | 21.83 | 49 == 40 | −1 | 1.20 | 65.56 |
| Mean | | | 2.49 | 240.72 | | | 1.27 | 94.33 |
| STD | | | 2.33 | 548.53 | | | 0.73 | 139.41 |

**Note:**
Location indicates the time-step in reference (R) and subject (S) curves that are most congruent for the severe impairment group. T, b, and c are Procrustean values that quantify rotation/reflection, scaling, and translation differences, respectively, between these segments of the reference and subject curves.

## ACKNOWLEDGEMENTS

We are indebted to all who participated in this study. We would like to thank Dr. Rachael Harrington and Dr. Evan Chan for guiding experiment design and assisting with data collection, Dr. Kathryn Laskey for guiding and reviewing the statistical analyses, and Dr. Qi Wei and Dr. Joseph Majdi for editing and reviewing this article. Finally, we are grateful to the MedStar National Rehabilitation Hospital research department for facilitating recruitment for this project.

### Funding

This work was supported by a Summer Fellowship and Dissertation Completion Grant provided by George Mason University. The funders had no role in study design, data collection and analysis, decision to publish, or preparation of the manuscript.

### Grant Disclosures

The following grant information was disclosed by the authors:
Summer Fellowship and Dissertation Completion Grant provided by George Mason University.

### Competing Interests

The authors declare that they have no competing interests.

## Author Contributions

- Khadija F. Zaidi conceived and designed the experiments, performed the experiments, analyzed the data, prepared figures and/or tables, authored or reviewed drafts of the article, and approved the final draft.
- Michelle Harris-Love conceived and designed the experiments, authored or reviewed drafts of the article, acquisition of IRB approval, recruitment, and approved the final draft.

## Human Ethics

The following information was supplied relating to ethical approvals (*i.e.*, approving body and any reference numbers):

This study received IRB approval through the Medstar Rehabilitation Hospital IRB, protocol number (947339-3).

## Data Availability

The datasets and MATLAB codes used in this study are available at Zenodo: Zaidi, Khadija F. (2023). Upper extremity kinematics: Development of a quantitative measure of impairment severity and dissimilarity after stroke. https://doi.org/10.5281/zenodo.7968664.

## Supplemental Information

Supplemental information for this article can be found online at http://dx.doi.org/10.7717/peerj.16374#supplemental-information.

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
