# Peer review of "Upper extremity kinematics: development of a quantitative measure of impairment severity and dissimilarity after stroke"

_PeerJ, doi:10.7717/peerj.16374_

## Round 0.1 · original submission · Major Revisions

Dear Dr. Zaidi,

Your manuscript titled "Upper extremity kinematics: Development of a quantitative measure of impairment severity and dissimilarity after stroke" was reviewed by two expert reviewers and based on their opinions and my review, The decision is “major revision”.

I agree with reviewer #1 that the discussion must be thoroughly revised. Currently it is devoid of any discussion of the literature and how current results agree or disagree with previous studies.

Also, please review your figures and figure legends to improve readability. This was also noted by the reviewers. In many of the figures the text is so small and very hard to read (e.g., Figures 1 & 3). Furthermore, figures 4 & 6 are upside down.

In addition, some of the information in the figures is not explained in the legends. For example – Figure 3: what is the meaning of the red line? (Peak velocity?)
Figure 4: what is the meaning of the blue and red lines? (red = control and blue = subject?)

Figure 6: The colors of control vs. subject (blue vs. red) seem to be reversed from figure 4. Why? This is confusing.

Figure 6: Some text seems to be covered in the figure (e.g., a partial letter appears before “right reach path, paretic right…”).

Figure 6: what is the difference between each pair of plots in each of the 6 parts of the figure. The plot title, and x- and y-axis are labeled the same, and the subject plot (in red) is the same. But the reference plot (in blue) is different. This should be explained clearly in the legend. The black connecting lines between the red and blue curves are also missing an explanation in the legend.

Reviewer 1 ·

Basic reporting

1.1. The article is well-written in clear and professional English.
1.2. Literature review in introduction and background sections is thoroughly conducted.
1.3. Line 118: Please include statistical analysis between the participant groups and mention whether the parameters in the groups were/were not statistically significant along with their p-value.
1.4. Please include units for age, months, and URFM column headers in Table 1 and 2.
1.5. Figure 1: Please enlarge the font size of the axis labels of the shown graph and ensure all text is readable.
1.6. Line 157: Please state the full form of the acronym ‘IRED’.
1.7. Figure 1: Please state the full form of the acronym ‘NDI’
1.8. Line 166: ‘recording’ needs to be replaced with ‘recorded’
1.9. Table 4: Please mention that body part that the measures represent (i.e., the hand of participant)

Experimental design

2.1 Methods used in this study are thoroughly described.
2.2 Line 162: How many markers were used to capture hand movement using the motion capture system? Please also state or illustrate the placement of the markers.
2.3 Line 240: Please mention the significance level for statistical analysis.

Validity of the findings

3.1 Figure 3: Please ensure the font-size of all text in figures is large enough to be readable.
3.2 Line 266: Please also mention the values for frequency (or percent difference) here.
3.3 Line 272: Please explicitly state that the difference between the two groups’ time of peak velocity was not significant (p-value>0.5).
3.4 Table 8 and Table 9: Please mention what T, b, and c indicate within the table captions.

Additional comments

4.1 Please include values, or percentage differences wherever relevant throughout the discussion section. For example, statement on Line 363 indicating higher mean velocities.
4.2. The limitations of the proposed method, as well as general limitations of the study may be discussed at the end of the discussion section (expanding upon those mentioned in conclusion).
4.3. Overall, the discussion section needs major improvements, including comparison of the methods and findings with literature.

·

Basic reporting

In this manuscript, the authors investigate the relationship between reaching trajectories in patients with mild and severe impairments after suffering from stroke. The language used is clear and grammatically sound, except for this one place : Line 166: "recording" -> "recorded"

The figures, on the other hand, are not clear. The text on the figures is very small and hard to read. Moreover, some figures are formatted in the wrong orientation, see Figure 6. The text in Fig 1 (right panel) is blurry.

Experimental design

The experiment design is sound and the conclusions are reasonable. There are very few significant correlations so I would encourage the authors to rewrite the Results section to specifically highlight parts that are significant and tone down on parts that are not. This is a suggestion about the writing because the way the paper is currently written, it is very hard to read.

Validity of the findings

Explained in section 2

---

## Round 0.2 · accepted · Accept

Dear Dr. Zaidi and Harris-Love,

Thank you for submitting your revised manuscript titled "Upper extremity kinematics: Development of a quantitative measure of impairment severity and dissimilarity after stroke". After reading the revised manuscript and the reviewer’s comments (see attached) I’m happy to let you know that decision is “accept”.

Reviewer 1 ·

Basic reporting

1.1. Several updates have been made based on the comments provided in the earlier review. The improvements made are satisfactory and I appreciate the authors for considering my comments.
1.2. The first section in methods ‘inclusion and exclusion criteria’ can be a part (sub-section) of the participants section.

Experimental design

2.1 I have no further comments about the experimental design and I am satisfied with the overall quality of the experimentation conducted in this study.

Validity of the findings

3.1 The discussion section has been improved as compared to the previous version. I am happy with the changes made by the authors.
3.2 Recent studies have been exploring the effects of cognitive-motor dual tasks on body movement. If cognitive loading was included as a part of this study in combination with the movement assist, do the authors think that better insights may have been obtained. Probably some of the limitations mentioned here can be addressed in future studies. Following are some examples of studies related to this topic.
3.2.1 Rice, J., Corp, D. T., Swarowsky, A., Cahalin, L. P., Cabral, D. F., Nunez, C., Koch, S., Rundek, T., & Gomes-Osman, J. (2022). Greater Cognitive-Motor Interference in Individuals Post-Stroke during More Complex Motor Tasks. Journal of Neurologic Physical Therapy, 46(1)
3.2.2 Abdollahi, M., Kuber, P. M., Pierce, M., Cristales, K., Dombovy, M., Lalonde, J., & Rashedi, E. (2023). Motor-Cognitive Dual-Task Paradigm Affects Timed Up & Go ( TUG ) Test Outcomes in Stroke Survivors. 2023 11th International IEEE/EMBS Conference on Neural Engineering (NER), 1–4.
3.3 Provided the current limitations in the last paragraph of the discussion, I am curious about the fact that whether the proposed method can be applied to evaluate movement of other body regions, in other types of tasks, including tasks besides reaching that are used commonly in clinical tests (TUG, STS,10MWT) like walking or turning? Below are some studies that may be beneficial to the authors in relating the novelty of the study to similar efforts in the field of motion analysis of stroke patients:
3.3.1 Alt Murphy, M., & Häger, C. K. (2015). Kinematic analysis of the upper extremity after stroke–how far have we reached and what have we grasped? Physical Therapy Reviews, 20(3), 137–155.
3.3.2 Abdollahi, M., Kuber, P. M., Shiraishi, M., Soangra, R., & Rashedi, E. (2022). Kinematic Analysis of 360◦ Turning in Stroke Survivors Using Wearable Motion Sensors. Sensors.
3.3.3 Mao, Y. R., Wu, X. Q., Li Zhao, J., Lo, W. L. A., Chen, L., Ding, M. H., Xu, Z. Q., Bian, R. H., Huang, D. F., & Li, L. (2018). The crucial changes of sit-to-stand phases in subacute stroke survivors identified by movement decomposition analysis. Frontiers in Neurology, 9(MAR).